# In-situ Cu Coating on Steel Surface after Oxidizing at High Temperature

**DOI:** 10.3390/ma12213536

**Published:** 2019-10-29

**Authors:** Na Li, Ruizhi Jia, Hongmei Zhang, Wei Sha, Yan Li, Zhengyi Jiang

**Affiliations:** 1School of Materials and Metallurgy, University of Science and Technology Liaoning, Anshan 114051, China; 17853481456@163.com (R.J.); lilyzhm68@163.com (H.Z.); 2State Key Laboratory of Metal Material for Marine Equipment and Application, Iron & Steel Research Institutes of An steel Group Corporation, Anshan 114009, China; 2323liyan@sina.com; 3School of Natural and Built Environment, Queen’s University Belfast, Belfast BT9 5AG, UK; W.Sha@qub.ac.uk; 4School of Mechanical, Materials, Mechatronic and Biomedical Engineering, University of Wollongong, Wollongong NSW 2522, Australia

**Keywords:** copper, oxidation, surfaces, composite materials

## Abstract

Almost all copper in scrap steel is recovered, so research on copper-bearing steel has profound practical significance. The surface enrichment of copper occurs in all copper-bearing steels studied in this paper after being heated at high temperature. In-situ oxidation-induced copper coatings were discovered on the descaled copper-bearing steels after heating at around 1150 °C for 2 h in air. Scattered copper precipitates in or under rust after heating at a lower temperature. A new concept was created using in-situ composites prepared by direct oxidation of matrix, and there was no bonding problem found between the coating and the matrix. The enrichment form of copper in steel is related to the oxidation rate, oxidation time, heating temperature and copper content.

## 1. Introduction

With the continuous development of the iron and steel industry, the amount of scrap has increased rapidly. Since copper is difficult to remove in the steelmaking process, it will accumulate in steel and its content will continue to increase when scrap is used as a raw material. Copper has been used widely in weathering steels [1], biological steels [2], antibacterial stainless steels [3], and other steels [4,5,6] for its properties of improving corrosion resistance, strength, and antibacterial capability. Furthermore, with the development of composites and their preparation technologies, researchers have reported various steel-copper composites. These kinds of composites can achieve high electrical conductivity, the high corrosion resistance of copper and the high strength of steels with low cost [7,8]. A high application value of steel-copper composites can also be obtained from its good plasticity, weldability and magnetic characteristics [9]. Due to its health benefits and noble red color, copper is used in countless applications as base coatings, intermediate coatings and top coatings.

Many researchers have observed the phenomenon that copper is enriched between the steel surface and the oxide scale in copper-bearing steels during high temperature mechanical working processes [1,8,10,11]. Reflecting this phenomenon, in this study a novel in-situ Cu coating was obtained on the surfaces of copper-bearing steels after being heated and descaled, which not only made use of copper in steel, but also prepared composite materials with good interface binding. At the same time, a new method for the preparation of cladding composite materials was proposed. The coating layer is formed on the surface of the substrate by the components that are not easy to oxidize in the process of high-temperature oxidation.

## 2. Materials and Methods 

In this work, copper-containing steels were prepared in a lab-scale high-frequency vacuum induction furnace by adding different contents of copper into low carbon steel. The chemical compositions of the test alloys are listed in Table 1.

Samples with the size of about 10 × 10 × 10 mm^3^ were taken from each steel, heated in air to 1000 °C, 1100 °C and 1150 °C, respectively, in a normal furnace for 2 h, followed by furnace cooling.

The samples were mounted with resin, ground, polished, but not etched. The microstructures of the cross sections were observed using a scanning electron microscope (SEM JSM-6500, JEOL, Tokyo, Japan) equipped with energy dispersive X-ray spectroscopy (EDS, Oxford, Abingdon, UK). X-ray diffraction (XRD X’Pert Powder, PANalytical, Almelo, Netherlands) was used to determine the phase compositions of the sample surface.

After heating at high temperature, the steel surfaces were treated with rust removal. The rust solvent was 30 g of citric acid and 50 g of sodium dihydrogen phosphate anhydrous, dissolved in 1 L of deionized water. The pH value of the descaling reagent measured by digital pH tester (MesuLab, Guangzhou, China) was about 1.6 and the descaling process lasted 24 h at room temperature.

## 3. Results

### 3.1. Heating and Oxidation

Figure 1 shows the back scattered electron (BSE) morphology and EDX surface scan of steel 1 after being heated at 1000 °C and 1150 °C, respectively. High temperature oxidation promotes the local enrichment of copper on the steel surface and in/under rust. Iron oxides appear in black shown in Figure 1a. After being heated at 1000 °C and 1100 °C, all samples showed scattered copper particles distributed near but not connected to the surface of the substrate, as shown in Figure 1a. These copper particles cannot effectively prevent further oxidation of steel substrate.

Figure 1b shows an integrated copper layer between the rusts and steel substrate after being heated at 1150 °C for 2 h. The copper layer connected perfectly to the substrate and there were no splits between them. The thickness of the copper coating changed from several microns to tens of microns. It can be deduced that further corrosion of steel substrate will be inhibited due to the presence of the continuous copper layer.

The EDX surface scans of Fe, Cu and O of Figure 1b show that oxygen is distributed mainly in the rust layer, which is the outermost layer of the sample, and the oxides are mostly iron oxides. The distribution of copper matches well with the BSE morphology where there is a little copper in the outer rust layer, and there is no phenomenon of copper diffusing from the inner substrate. Therefore, the aggregated copper comes from the oxidized substrate surface, which then becomes the later rust layer. Copper segregates during the oxidizing process, and the quantity of aggregated copper is related to the thickness of oxidized substrate and copper content in the substrate.

It can be seen in Figure 1b that the copper layers through with grain boundaries can easily enter the substrate, which can cause hot shortness if the steel is subjected to a stress. Therefore, an alternative hot processing route should be explored. Local heating near the surfaces of near-final products is preferred as well. 

Figure 2 shows the SEM image of the cross section of steel 2 after being heated at 1000 °C, 1100 °C, and 1150 °C for 2 h. The oxides of iron formed a thick rust layer (some of the rust fell off during sample preparation), and the segregation form of copper on steel surface is the same as that of steel 1, as shown in Figure 2. Due to the high copper content in steel 2, the copper-rich phase precipitates in network and granular form in iron matrix. Meanwhile, there is no copper-rich precipitation in steel 1, with low copper content, as shown in Figure 1b2. Therefore, it is likely that a certain amount of copper was dissolved in the steel as a solid solution uniformly. EDX results show that the compositions of precipitated phase in the substitute and the copper-rich phase on the surface were similar.

### 3.2. Descaling

The macrographs of steel 2 before and after descaling are shown in Figure 3. For the steels with integrate copper coating, descaling treatment was conducted to remove the rusts from the test steels while maintaining the copper layers. Because of the noble color of copper, it can be judged with the naked eye.

Figure 4 shows the XRD patterns on the surfaces of steels 1 and 2 after descaling. XRD test results show that the surfaces of the descaled steels were made up of pure copper and pure iron, and that pure copper was the main ingredient. The copper content in bulk steel 1 was much lower than that in steel 2, while the composition of the descaled surface was similar. In other words, the descaled surface of steel 1 was also covered mainly with copper. There was no ferric oxide identified by XRD.

The surface microstructures of the experimental steel after descaling were observed with SEM and BSE, as shown in Figure 5. Most parts of the surface were covered with nearly pure copper, as shown in the XRD results. However, there were some ferric oxides mixed with pure copper left on the descaled surface according to the EDX results, as shown in Figure 5b. Although the ferric oxides with shapes approximating blocks or plates appear on the upper layer of the copper coating, the quantity of ferric oxides is not high enough to be identified by XRD, according to Figure 4.

## 4. Discussion

### 4.1. The Movement of Copper Atoms

It is well known that the oxidation rate of copper is relatively low compared to that of iron. In the process of high temperature oxidation, oxygen atoms prefer to combine with iron atoms to form non-metallic oxidation products such as FeO, Fe_2_O_3_ and Fe_3_O_4_. In this process, the volume of the oxide layer expands with an increase of material weight, and the distance between the oxide layer and the metal atoms in the matrix increases. This is because the bonding force between oxide molecules and the metal atoms is weakened after oxidation, and usually accompanied by some defects [12].

The bonding force between metallic atoms is stronger than that between the metal and oxide atoms. With the oxidation of the steel surface, the bonding force between the unoxidized copper atoms and the surrounding oxide molecules weakens, while the bonding force between copper atoms in the oxide layer and the adjacent iron (and/or copper) atoms in the matrix is strengthened even with increasing distance, as shown in Figure 6. Therefore, during oxidation, the unoxidized copper atoms move continuously from the oxide layer to and remain on the surface of the metal substrate due to the attraction between metal atoms.

In the oxidation process, iron atoms in lattice positions on the substrate surface combine with oxygen and deviate from the initial positions to form iron oxides. The migration of surrounding iron atoms and the increase in the distance between the metal atoms and non-metal molecules is favorable for the directional movement of copper atoms. Under certain conditions, oxidation and copper atom migration occur simultaneously on the steel surface. When the two velocities match, a continuous copper enrichment layer can be formed over a period of time, and a multi-layer structure composed of the outer rust layer, copper layer, and substrate can be formed on the steel surface. Therefore, the copper layer in this structure originates from the movement of copper atoms in the oxide layer. Macroscopically, this “copper coating” is formed by the substrate itself. The migration of copper atoms from the oxide layer to the substrate surface is the result of metal bonding. Therefore, there is no binding problem between the copper layer and the substrate.

### 4.2. Influence Factors

The formation of the copper cladding structure may be influenced by the following factors: oxidation temperature, oxidation rate, oxidation time, and copper content in the steel.

The oxidation rate should correspond to the moving speed of the copper atoms on the oxidation interface, although these two values are difficult to measure accurately at present. If the oxidation rate is high, the iron atoms around the copper atoms are rapidly oxidized. The copper atoms are far away from the unoxidized metal atoms on the substrate surface, so they lose the driving force and cannot move to the interface in time to form a copper layer. On the other hand, if the oxidation rate is low, the moving speed of the copper atoms to the metal matrix will also be very low. In this process, some copper atoms themselves maybe oxidized, and there is no reason for the copper atoms inside the metal matrix to move toward the surface. Therefore, no copper layer will be formed in this case.

At the same time, the rate at which copper atoms move and iron atoms oxidizes is also related to temperature. When the heating temperature is 1000 °C, copper atoms attract each other, but fail to bond well with the substrate, indicating that the interaction force between copper atoms is greater than that between the copper and iron atoms, and the oxidation rate is still relatively high under this condition. The higher the temperature, the faster the atoms move and oxidize. However, these two speeds should match each other, not the faster the better.

The formation of a copper cladding layer is also related to the copper content in steel. As mentioned above, the formation of the copper layer is a process of continuous movement and accumulation of copper atoms in the oxidized layer towards the substrate surface. Therefore, when the copper content is low, it takes a long oxidation time and a thick oxide layer to form a continuous copper layer, and the opposite is true when the copper content is high.

## 5. Conclusions

In this paper, by observing the segregation behavior of copper on a steel surface after heating at high temperature, an in-situ copper-clad steel composite material was prepared, which could not only utilize copper in steel, but also prepare composite material with good interface bonding. The following conclusions can be made:
(1)After high temperature heating, copper was segregated on the surface of copper bearing steel, which comes from the oxidized steel surface. When the temperature was 1150 °C, a complete copper coating was formed on the steel surface.(2)After removing the rust, the copper coating was uncovered on the steel surface. The in-situ copper coating was generated from the steel matrix, and there was no bonding problem between them.(3)A lower oxidation rate, longer oxidation time, appropriate oxidation temperature, and higher copper content were favorable for the formation of a copper-enriched layer. Further investigation is needed to find appropriate relationships between the various operating factors and the characteristics of the copper layer.

## Figures and Tables

**Figure 1 materials-12-03536-f001:**
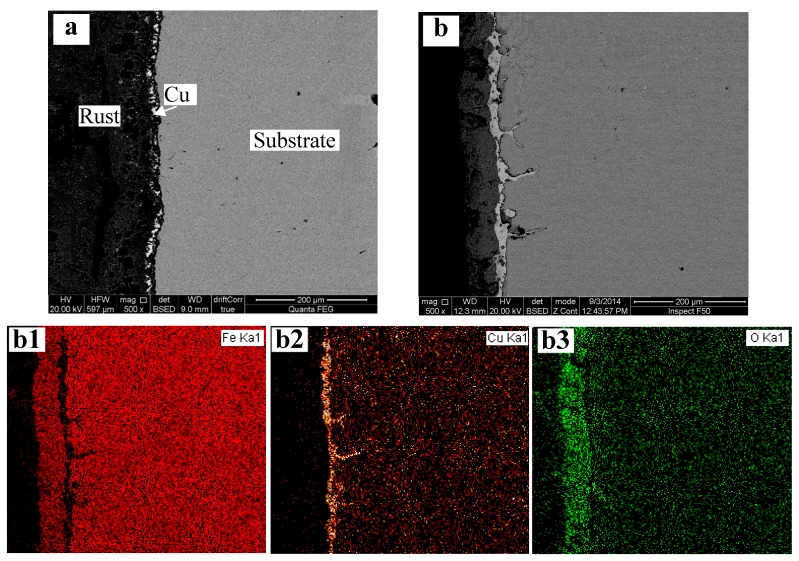
Back scattered electron (BSE) morphology of steel 1 after heading at 1000 °C (**a**) and 1150 °C (**b**) for 2 h and EDX surface scan of Fe (**b1**), Cu (**b2**) and O (**b3**) in (**b**).

**Figure 2 materials-12-03536-f002:**
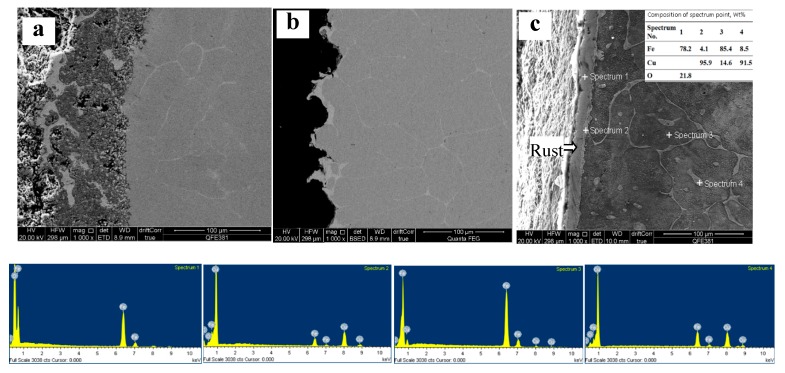
Scanning electron microscope (SEM) (BSE) morphologies on the cross section of steel 2 after being heated at 1000 °C (**a**), 1100 °C (**b**) and 1150 °C (**c**) for 2 h, and EDX results of corresponding points in (**c**).

**Figure 3 materials-12-03536-f003:**
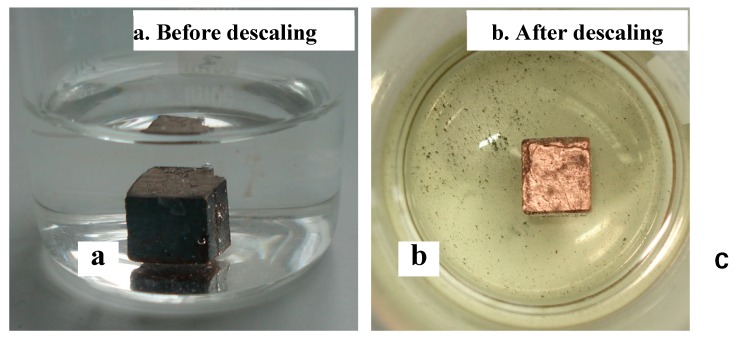
Macrographs of heated steel 2 before (**a**) and after (**b**) descaling in a beaker.

**Figure 4 materials-12-03536-f004:**
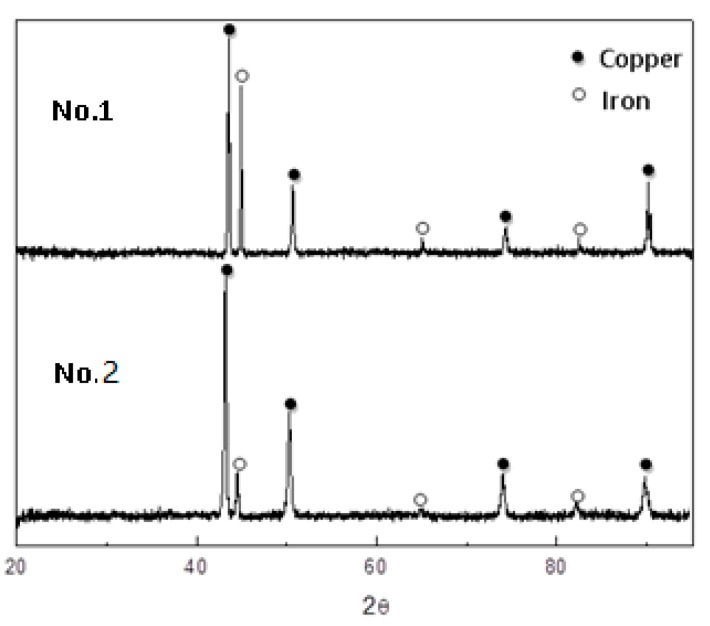
X-ray diffraction (XRD) patterns on the surface of descaled steels 1 and 2.

**Figure 5 materials-12-03536-f005:**
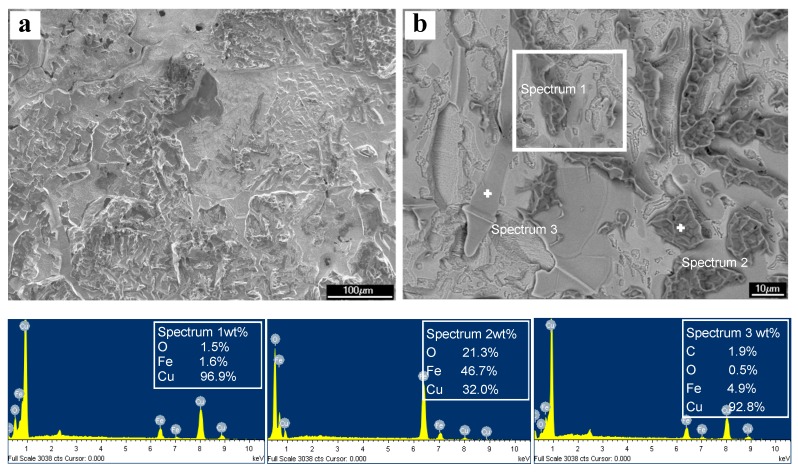
SEM (**a**) and BSE (**b**) micrographs and corresponding EDX spectra on the surface of steel 2 after descaling.

**Figure 6 materials-12-03536-f006:**
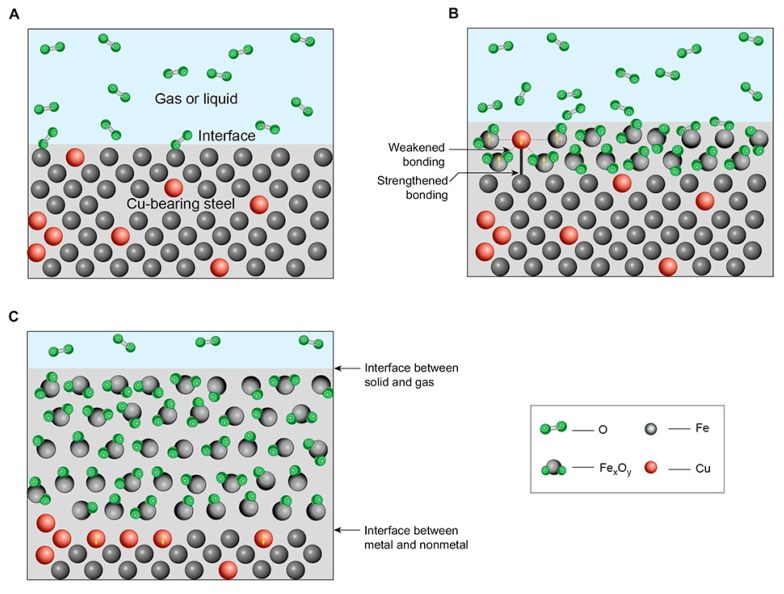
Schematic diagram of copper enrichment mechanism.

**Table 1 materials-12-03536-t001:** Compositions of test steels (mass %).

Steel No.	C	Cu	Si	Mn	P	S	Fe
1	0.001	2.38	0.018	0.035	0.010	0.007	Bal
2	0.001	12.64	0.056	0.064	0.009	0.007	Bal

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
