# Peer review of "In-situ Cu Coating on Steel Surface after Oxidizing at High Temperature"

_materials, 2019, doi:10.3390/ma12213536_

Round 1
Reviewer 1 Report
Dear authors,
thank you for the interesting work.
I have just a few comments:
line 48 lease check the specimen size.
Can you derive a diagram showing the cu sheet-thickness over the process conditions, like the heating temperature?
Please add a few words of summary to the conclusion.
Regards
a Reviewer
Author Response
Author’s reply to the review report (Reviewer 1):
line 48 lease check the specimen size.
The specimen size should be about 10×10×10 mm3, thanks for the comment.
Can you derive a diagram showing the cu sheet-thickness over the process conditions, like the heating temperature?
In this study, no continuous copper coating was formed in the steels heated at a lower temperature, such as 1000℃and 1100℃.
There is a complete copper sheet was formed on the steel surface after heating at 1150℃ and the thickness of the copper sheet changes from several microns to tens of microns as shown in Fig. 1b.
Therefore, it is difficult to derive a diagram showing the copper sheet-thickness over heating temperature.
As for the influence of heating duration on the thickness of copper sheet, if the heating duration is prolonged at high temperature, the continuous copper coating layer will prevent the continuous oxidation of steel surface, and the copper on the surface will also be partially oxidized.
Please add a few words of summary to the conclusion.
Conclusions were modified to be:
In this paper, by observing the segregation behavior copper on the steel surface after heating at high temperature, the in-situ copper-clad steel composite material was prepared, which could not only utilize copper in steel, but also prepared composite material with good interface bonding. The following conclusions can be made:
After high temperature heating, copper segregates on the surface of copper bearing steel, which comes from the oxidized steel surface. When the temperature is 1150℃, a complete copper coating is formed on the steel surface. After removing the rust, the copper coating was uncovered on the steel surface. The in-situ copper coating is generated from the steel matrix, and there is no bonding problem between them. A lower oxidation rate, longer oxidation time, appropriate oxidation temperature and higher copper content were favorable for the formation of a copper enriched layer.
In addition, the introduction, methods and some results were improved, respectively. All modifications were marked in the text.

Reviewer 2 Report
Comments to the Author:
The author of this paper present an interesting preparation of in-situ Cu coating on the surfaces of copper-bearing steels after being heated and descaled. Nevertheless, some details should be considered by the authors:
Introduction
COMMENT: Page 1, line 43: This new method should be briefly described at the end of the introduction.
Results and discussion
COMMENT: Page 2, Fig. 1a and b: For a direct comparison, the magnification of both images presented in Fig 1a and b should be the same. Therefore, I suggest the authors to change Fig. 1, accordingly.
COMMENT: Page 3, Fig. 2: For a direct comparison, I suggest the authors to add the SEM images of steel 1 after being heated at 1000 and 1100 oC. Although the EDX data are shown, the EDX spectra may also be added.
The reported data are discussed and commented and the results support the authors conclusions. Therefore, I think that this paper is suitable for publication.
Author Response
Author’s reply to the review report (Reviewer 2):
The author of this paper present an interesting preparation of in-situ Cu coating on the surfaces of copper-bearing steels after being heated and descaled. Nevertheless, some details should be considered by the authors:
Introduction
COMMENT:Page 1, line 43: This new method should be briefly described at the end of the introduction.
The following introduction content was added, as shown in page 1, lines 44 and 45:
At the same time, a new method for the preparation of cladding composite materials was proposed. The coating layer is formed on the surface of the substrate by the components that are not easy to oxidize in the process of high-temperature oxidation.
Results and discussion
COMMENT: Page 2, Fig. 1a and b: For a direct comparison, the magnification of both images presented in Fig 1a and b should be the same. Therefore, I suggest the authors to change Fig. 1, accordingly.
Fig. 1a has been changed according to this suggestion, thanks.
COMMENT: Page 3, Fig. 2: For a direct comparison, I suggest the authors to add the SEM images of steel 1 after being heated at 1000 and 1100 oC. Although the EDX data are shown, the EDX spectra may also be added.
We think the reviewer meant adding the SEM images of steel 2 after being heated at 1000 and 1100 oC, because the current SEM image is the image of steel 2 heated at 1150 oC. The suggested figures and EDX spectra have been added.
The relevant situations of steel 1 were shown in Fig. 1.
The reported data are discussed and commented and the results support the authors conclusions. Therefore, I think that this paper is suitable for publication.
In addition, the introduction, methods and some results were improved, respectively. All modifications were marked in the text. Thanks!

Reviewer 3 Report
The article has a high scientific interest. The study is well planned, and has a high scientific interest. The images obtained in the SEM microscope, are of great quality, and greatly facilitate the interpretation of the results. The graphics are clear and well explained, the results being very interesting. The conclusions, are a bit poor, should be reviewed and completed. In general it seems an interesting article. It has sufficient quality to be published.
Author Response
Authors reply:
Conclusions were modified to be:
In this paper, by observing the segregation behavior copper on the steel surface after heating at high temperature, the in-situ copper-clad steel composite material was prepared, which could not only utilize copper in steel, but also prepared composite material with good interface bonding.The following conclusions can be made:
After high temperature heating, copper segregates on the surface of copper bearing steel, which comes from the oxidized steel surface. When the temperature is 1150℃, a complete copper coating is formed on the steel surface. After removing the rust, the copper coating was uncovered on the steel surface. The in-situ copper coating is generated from the steel matrix, and there is no bonding problem between them. A lower oxidation rate, longer oxidation time, appropriate oxidation temperature and higher copper content were favorable for the formation of a copper enriched layer.
In addition, the introduction, methods and some results were improved, respectively. All modifications were marked in the text. Thanks!
